# Calcium Signaling and the Response to Heat Shock in Crop Plants

**DOI:** 10.3390/ijms25010324

**Published:** 2023-12-26

**Authors:** Xinmiao Kang, Liqun Zhao, Xiaotong Liu

**Affiliations:** Key Laboratory of Molecular and Cellular Biology of Ministry of Education, Hebei Research Center of the Basic Discipline of Cell Biology, Hebei Collaboration Innovation Center for Cell Signaling, Hebei Key Laboratory of Molecular and Cellular Biology, College of Life Sciences, Hebei Normal University, Shijiazhuang 050024, China; kangmiaomiao1314@163.com

**Keywords:** crop yield, thermotolerance, Ca^2+^ signaling

## Abstract

Climate change and the increasing frequency of high temperature (HT) events are significant threats to global crop yields. To address this, a comprehensive understanding of how plants respond to heat shock (HS) is essential. Signaling pathways involving calcium (Ca^2+^), a versatile second messenger in plants, encode information through temporal and spatial variations in ion concentration. Ca^2+^ is detected by Ca^2+^-sensing effectors, including channels and binding proteins, which trigger specific cellular responses. At elevated temperatures, the cytosolic concentration of Ca^2+^ in plant cells increases rapidly, making Ca^2+^ signals the earliest response to HS. In this review, we discuss the crucial role of Ca^2+^ signaling in raising plant thermotolerance, and we explore its multifaceted contributions to various aspects of the plant HS response (HSR).

## 1. Ca^2+^ Signaling and Plant Thermotolerance

The rapid rise in CO_2_ levels on Earth is causing global warming, leading to more frequent extreme high-temperature (HT) events. Heat is a major factor in reducing crop yields. It disrupts homeostasis, affects seed germination, and alters (and ultimately stunts) plant growth [1]. Without sufficient adaptation, genetic enhancements, and fertilization, every one degree Celsius increase in global mean temperature could lead to significant reductions in global wheat, rice, and maize yields (averaging 6.0%, 3.2%, and 7.4%, respectively) [2]. Therefore, understanding the molecular mechanisms underlying plant responses to HT stress is vital for improving agricultural production and ensuring future food security.

HT can elevate the intracellular concentration of calcium (Ca^2+^), a common second messenger in both animal and plant cells. Plants exposed to HT can experience heat shock (HS), which triggers a rise in cytosolic Ca^2+^ and disrupts the oscillations in Ca^2+^ levels [3]. Ca^2+^ plays a critical role in maintaining the normal physiological functions of plant cells and is involved in various physiological processes in plants. Additionally, as a ubiquitous second messenger, Ca^2+^ participates in plant responses to various stressors. Thus, understanding the impact of Ca^2+^ on plant thermotolerance is essential for the development of heat-resistant crops [4,5]. 

Ca^2+^ is a versatile intracellular signal; information is encoded based on temporal and spatial patterns of Ca^2+^ concentration changes. These patterns are decoded by Ca^2+^-sensing effectors such as Ca^2+^-permeable channels and Ca^2+^-binding proteins to initiate specific cellular responses [5]. The induction of the Ca^2+^ signal represents the most rapid response to elevated temperatures in plants. In one study, the cytosolic Ca^2+^ concentration in wheat peaked within 10–15 min of a sudden temperature increase from 24 to 36 °C before returning gradually to baseline as the HS response (HSR) continued [6].

Ca^2+^ entry into the cytoplasm is facilitated by several families of protein channels, including cyclic nucleotide-gated channels (CNGCs), glutamate (Glu) receptor-like channels (GLRs), annexins, and mechanosensitive (MS) channels. Each of these channel types plays a crucial role in promoting an influx of Ca^2+^. Furthermore, plants possess various Ca^2+^-binding proteins that decode and transmit the primary Ca^2+^ signal to elicit specific cellular responses. These proteins include calmodulins (CaMs), CaM-like proteins (CMLs), Ca^2+^-dependent protein kinases (CDPKs or CPKs), Ca^2+^- and CaM-dependent protein kinases (CCaMKs), calcineurin B-like proteins (CBLs), and CBL-interacting protein kinases (CIPKs). When these proteins bind Ca^2+^, they undergo conformational changes that allow them to initiate downstream signaling events via interactions with other proteins or molecules. This mechanism enables plants to translate variations in the intracellular Ca^2+^ concentrations into specific cellular responses, including changes in gene expression, enzyme activity, or ion channel activity [7,8,9]. 

Recent studies have focused on understanding how plants detect Ca^2+^ increases due to HS, and many studies have identified crucial molecules and signaling pathways involved. Here, we review the latest findings on thermosensing in different crop species.

## 2. Ca^2+^-Permeable Channels Perceive Elevated Temperatures

Presently, the identity of a definitive thermosensor in plants remains elusive. It is hypothesized that Ca^2+^ channels located in the plasma membrane (PM) are crucial players in perceiving elevated temperatures. These channels could be activated directly or indirectly, leading to increased cytoplasmic Ca^2+^ levels. Consequently, thermosensors may function as Ca^2+^ channels, directly modulating Ca^2+^ signaling in response to external stimuli. Alternatively, they may act as regulatory elements that influence the membrane lipid composition, which is closely linked to Ca^2+^ channel activity. Another possibility is that they operate as GLRs, initiating an influx of Ca^2+^ in response to external stimuli.

Genome sequencing has revealed that plants lack typical animal Ca^2+^ channels such as voltage-dependent Ca^2+^ channels, transient receptor potential channels, purinergic P2X receptor channels, and cysteine loop channels. Instead, they have expanded families of CNGCs, GLRs, annexins, reduced hyperosmolarity-induced [Ca^2+^] increase channels (OSCAs), “Mid1-complementing activity” channels (MCAs), two-pore channels (TPCs), MS-like channels (MSLs), and Piezo channels (MSPs). In the context of HSR, we will explore several potential candidates among these Ca^2+^-permeable ion channels (Figure 1).

### 2.1. Heat Sensing via CNGCs

CNGCs are non-specific cation channels that regulate the flow of Ca^2+^ by binding to ligands such as cAMP and cGMP. CNGCs are activated in response to various abiotic and biotic stresses [10,11]. Plant and animal CNGCs belong to the K^+^-selective shaker channel family and share similarities in their amino acid sequences and overall structures [12]. They possess cyclic nucleotide-binding domains and one or more CaM-binding domains at their cytosolic N- and C-termini [13,14]. 

HS induces PM fluidization, allowing an influx of Ca^2+^ into the cytoplasm. Furthermore, nucleotide cyclases can elevate cAMP and cGMP levels under HS conditions, further promoting the influx of Ca^2+^ and activating associated signaling pathways. CNGCs were initially discovered in barley [15]; subsequent studies identified 20 and 16 CNGCs in the genomes of *Arabidopsis* and rice, respectively [16,17]. Currently, plant CNGCs are known to participate in a range of biological processes [18,19,20,21]. 

The involvement of CNGCs in the plant HSR was initially suggested through studies of *Arabidopsis* CNGC2 and *Physcomitrella patens* CNGCb null mutants. These studies revealed that a loss of CNGC2/CNGCb function enhanced thermotolerance at the seedling stage, accompanied by a moderate increase in the accumulation of heat-responsive proteins compared with wild-type plants [21,22]. Intriguingly, AtCNGC2 dysfunction has been linked to thermosensitivity at the reproductive stage, indicating a developmental stage-dependent role in the HSR [23]. Recent research also revealed differential regulation of AtCNGC4 in shoots and roots under variable temperature conditions [24]. The Ca^2+^ current in *Atcngc6* was lower than in the WT. However, the Ca^2+^ current in *AtCNGC6* overexpression plants is significantly higher than in WT. The study provided evidence that CNGC6 mediates heat-induced Ca^2+^ influx and enhances the expression of genes encoding HS proteins (HSPs) [25]. Nitric oxide (NO) and hydrogen peroxide (H_2_O_2_) have been identified as downstream partners of CNGC6 that contribute to increased thermotolerance [26,27].

In rice, *oscngc14* and *oscngc16* mutant plants exhibited reduced survival rates, elevated levels of H_2_O_2_, and increased cell death under HS conditions. These findings indicate that OsCNGC14 and OsCNGC16 modulate thermotolerance by regulating cytosolic Ca^2+^ levels in response to HS, underscoring their critical roles in the HSR in rice [28]. Interestingly, of the 16 *CNGC* genes in rice, 10 were notably upregulated under low-temperature conditions, warranting further investigation into their expression changes in response to HS. Additionally, *CNGC* genes from other plants (e.g., *P. patens*, *Nicotiana tabacum*, *Brassica oleracea*, and *Mangifera indica*) have been suggested to contribute to thermotolerance, indicating their potential importance in dealing with extreme temperature conditions [29].

### 2.2. Heat Sensing via GLRs

In mammals, ionotropic Glu receptor cation channels (iGluRs) respond to the neurotransmitter Glu, initiating Ca^2+^ signal cascades [30]. The first evidence of Glu signaling in plants was uncovered by Lam et al. [31], who identified the *AtGLR* gene family in *Arabidopsis*. Through sequence analysis and comparisons, homologous Glu receptor variants have been identified in dicot and monocot species [32,33,34]. The *Arabidopsis* genome contains 20 GLRs, while the rice and poplar genomes harbor 13 and 61 GLRs, respectively [35,36]. Plant GLRs display significant sequence and structural homologies with animal iGluRs, including high degrees of amino acid sequence identity in their channel and ligand-binding domains [37].

Studies on the physiological roles of GLRs have revealed their involvement in diverse processes such as photosynthesis, carbon/nitrogen balance, Ca^2+^ regulation, root architecture, pollen tube growth, defense signaling, and environmental stress responses [38,39,40,41,42]. For example, GLR3.3 has been implicated in the response of *Arabidopsis* to pathogen infections [43], while GLR3.5 promotes drought tolerance in faba beans [44]. In tomatoes, GLR3.3 and GLR3.5 mediate cold stress tolerance by regulating apoplastic H_2_O_2_ production and redox homeostasis [45]. 

Although research on the involvement of *GLR* genes in plant thermotolerance is limited, some studies indicate a potential role in responding to HT stress. For instance, one study investigated the effect of Glu on the survival of maize seedlings under HS conditions and its impact on Ca^2+^ signaling [46]. The results showed that pretreatment with Glu enhanced the thermotolerance of maize seedlings, possibly through GLR-mediated Ca^2+^ signaling. Glu is emerging as a novel signaling molecule with involvement in a wide range of physiological processes in plants. Further exploration is needed to uncover the specific functions of GLRs, particularly in the context of thermotolerance. Additional research methodologies should be employed to elucidate the mechanisms by which GLRs contribute to thermotolerance in plants.

### 2.3. Heat Sensing via Annexins

Annexins, an evolutionarily conserved family of proteins present in a variety of organisms, are renowned for their ability to bind to Ca^2+^ and phospholipids. They play pivotal roles in a plethora of cellular processes [47,48]. Annexins exhibit peroxidase and ATPase/GTPase activities and are associated with the regulation of Ca^2+^ channels [49]. Their functions encompass a wide range of intracellular and extracellular phenomena, including vesicular trafficking, organization of the membrane–cytoskeleton, exocytosis, endocytosis, phagocytosis, ion channel regulation, and apoptosis [50]. The genomes of *Arabidopsis*, rice, and wheat contain 8, 10, and 25 annexin genes, respectively [51,52,53]. Annexins feature a conserved protein core domain capable of binding both Ca^2+^ and phospholipids alongside an N-terminal domain that varies in sequence and length among different annexins [54].

In plants, annexins participate in environmental stress responses, and they play roles in growth, development, and signaling [55]. Some plant annexins regulate the level of free cytosolic Ca^2+^, and certain annexins can form Ca^2+^-permeable channels in lipid bilayers or vesicles [56]. Moreover, plant annexins may possess peroxidase activity or ATPase/GTPase activity, each contributing to functional specificity [57]. Annexins play a broad regulatory role in diverse biochemical and cellular processes, including Ca^2+^ channel regulation, and in plant growth, development, and biotic and abiotic environmental stress responses [57]. 

Several studies have highlighted the pivotal role of annexins in plant thermotolerance. For example, NnANN1, a heat-induced annexin, was identified in the embryonic axes of the sacred lotus (*Nelumbo nucifera* Gaertn.) through comparative proteomics. NnANN1 expression increased substantially in response to HT treatment. Ectopic expression of NnANN1 in *Arabidopsis* enhanced the thermotolerance of transgenic seeds. These seeds exhibited increased peroxidase activity, reduced lipid peroxidation, and reduced reactive oxygen species (ROS) levels compared with wild-type seeds [58]. Among crop plants, the soybean annexin GmANN has been found to promote thermotolerance and humidity and to improve seed vigor. *GmANN*-transgenic *Arabidopsis* seeds displayed enhanced heat resistance and greater seed vitality under HT stress and high humidity compared with wild-type seeds [59]. Additionally, *GmANN* overexpression in plants led to increased peroxidase activity, decreased lipid peroxidation, and reduced ROS levels compared with wild-type plants. OsANN1, a rice annexin, has been reported to enhance abiotic stress tolerance by modulating antioxidant accumulation. Overexpression of OsANN1 promoted the activities of superoxide dismutase (SOD) and catalase (CAT), which regulate the H_2_O_2_ content and redox homeostasis in cells. This suggests the existence of a feedback loop that controls OsANN1 and H_2_O_2_ production under conditions of abiotic stress. Elevated *OsANN1* expression conferred cellular protection against HS and H_2_O_2_, with significant cytosolic localization observed following HT treatment [60]. In another study, phylogenetic analysis revealed that radish annexins (RsANNs), along with *Arabidopsis* and rice annexins, clustered into five groups with similar motif patterns. Real-time quantitative PCR showed that most *RsANN* genes are induced by such abiotic stressors as heat, drought, salinity, oxidation, and abscisic acid (ABA). Furthermore, overexpression of *RsANN1a* enhanced the growth and thermotolerance of HS-treated Arabidopsis plants, while the knockdown of RsANN1a using artificial microRNAs resulted in decreased survival [61].

Annexins constitute a diverse multigene family with multifaceted roles in plants; however, our understanding of their functions is still in its infancy. Further research is needed to delve into the potential mechanisms by which annexin genes contribute to thermotolerance in plants.

### 2.4. Heat Sensing via OSCAs

Recently discovered MS Ca^2+^ channels called OSCAs have the ability to detect and respond to changes in osmotic pressure, regardless of whether it originates externally or internally. OSCAs are essential for regulating the flow of Ca^2+^ in plants; thus, they play a crucial role in plant growth and adaptation to environmental stress. OSCA1, the first OSCA identified in *Arabidopsis thaliana*, is responsible for raising cytosolic Ca^2+^ levels by triggering an influx of Ca^2+^. In fact, this channel is responsible for the increase in intracellular Ca^2+^ triggered by multiple stimuli in plants [62]. In total, 15, 11, 12, and 62 *OSCA* genes have been identified in *Arabidopsis* [62], rice [63], maize [64], and cotton [65], respectively. 

In rice, 11 *OsOSCA* genes were identified from the *Oryza sativa* L. *Japonica* genome database. These genes were categorized into four clades based on their sequence composition and phylogenetic relationships. Their expression profiles were assessed at different stages of vegetative growth, during reproductive development, and in response to osmotic stress. Remarkably, differential expression due to osmotic stress was noted for 10 of the 11 genes [63]. 

Similarly, in maize, a comprehensive analysis of the *ZmOSCA* gene family was conducted through bioinformatics and expression studies. A total of 12 *OSCA* genes were identified from the maize genome database. These genes were classified into four groups (I–IV) based on their sequence composition and phylogenetic relationships. Notably, when maize plants were subjected to HS (40 °C), *OSCA* gene expression exhibited two distinct patterns. Six genes, *ZmOSCA1.4*, *ZmOSCA2.1*, *ZmOSCA2.2*, *ZmOSCA2.5*, *ZmOSCA3.1*, and *ZmOSCA4.1*, displayed a significant increase in expression after 1 h of HT stress, reached a peak, and then underwent rapid downregulation, followed by a gradual increase. Meanwhile, the expression patterns of *ZmOSCA1.3*, *ZmOSCA1.5*, and *ZmOSCA2.4* were contrary to those of all other maize *OSCA* family members [66]. These findings strongly suggest that *OSCA* genes play a crucial role in mediating plant responses to HS.

### 2.5. The Functions of MSLs, MCAs, and MSPs in Plants following HT Treatment

In addition to OSCAs, plants possess several other types of MS ion channels, including MSLs, MCAs, and MSPs. These channels are permeable to Ca^2+^ and play crucial roles in enabling plants to respond to mechanical stimuli and changes in osmotic pressure.

MSLs are MS ion channels found in both bacteria and plants. In *Arabidopsis*, MSL2 and MSL3 are essential to protect plastids against bursting inside leaf epidermal cells during growth under conditions of high osmotic pressure [67]. MSL8 responds to PM distortion during pollen grain rehydration and germination [68]. Additionally, MSL10 and MSL9 exhibit MS ion channel activity in root protoplasts [69]. In rice, most *OsMSL* genes are expressed during reproductive growth, suggesting their involvement in plant growth, development, and stress responses [70].

MCAs are plant-specific MS Ca^2+^-permeable channels found in such plants as *Arabidopsis*, rice, and *N. tabacum* [71,72]. In *Arabidopsis*, two such channels, MCA1 and MCA2, have been identified. MCA1 and MCA2 were shown to mediate a Ca^2+^ influx in yeast cells and to function in the rapid elevation of cytosolic Ca^2+^ levels during cold exposure [73]. wild-type *Arabidopsis* (Col-0) plants exhibited higher cold-induced cytosolic Ca^2+^ concentrations compared with *mca1* and *mca2* mutant plants. Conversely, the double mutant *mca1 mca2* showed freezing and chilling sensitivity, indicating that MCA1 and MCA2 confer cold stress tolerance to plants [74].

While MSPs are primarily found in animals, one gene encoding an MSP has been identified in *Arabidopsis* [75,76]. AtPiezo, an uncharacterized MSP in *Arabidopsis* and an ortholog of animal MSPs was recently discovered [77]. AtPiezo was necessary to inhibit a systemic infection by CMV-2aT∆2b or turnip mosaic virus tagged with a green fluorescent protein (TuMV-GFP). Viral infection induced *AtPiezo* expression, especially in the petioles of rosette leaves. This study marks the first demonstration of the biological function of an MSP in plants. Further, it suggests that this mechanism represents a common antiviral strategy since many monocots and dicots possess a single MSP ortholog.

While the direct involvement of these channels in the HSR has not been extensively studied, their potential role in such responses warrants further investigation. 

A list of Ca^2+^ channels involved in sensing heat has been included as Table 1.

## 3. Ca^2+^-Binding Protein Involvement in the HSR

Under HS conditions, the intracellular Ca^2+^ concentration in plants increases, primarily due to an influx of Ca^2+^ from outside the cell or the release of Ca^2+^ from intracellular stores [22,78]. This rise serves as a pivotal “signal” that triggers a complex signal transduction cascade. This signal, in conjunction with downstream effector proteins, including CaMs, CMLs, CBLs, and CDPKs/CPKs, serves as a molecular code that must be interpreted by the cell. Together, these proteins create a sophisticated signaling network that transmits stress signals and orchestrates a multitude of responses in plants [79]. In this section, we will concentrate on candidate Ca^2+^-binding proteins that were reported to be involved in the HSR in crops (Figure 1).

### 3.1. CaMs in HS Signaling

CaMs, highly conserved sensor proteins containing EF-hand motifs, are found in various plant organelles. In total, 9, 5, and 8 *CaM* genes have been identified in *Arabidopsis* [80,81], rice [82], and maize [83], respectively. CaMs are ubiquitously expressed in all eukaryotic cells and possess a dumbbell-shaped structure. They are multifunctional, consisting of two globular lobes at the N- and C-termini connected by a flexible central linker. Each lobe contains a pair of EF-hand motifs capable of binding Ca^2+^ ions with positive cooperativity [84]. The EF-hand motif in CaM is characterized by a helix–loop–helix (D-X-D) conformation, consisting of 12 amino acid residues, totaling 36 amino acid residues across both motifs. In the D-X-D motif, the 14th and 16th positions are fixed, while the 15th position can be occupied by any amino acid [85]. These motifs are involved in interactions with a diverse array of downstream target proteins, including ion channels, pumps, antiporters, kinases, phosphatases, transcription factors, and enzymes involved in metabolic pathways. *Arabidopsis* CaM isoforms CaM1/4, CaM2/3/5, CaM6, and CaM7 were found to bind to CNGC6 to varying degrees, and this binding was dependent on the presence of Ca^2+^ and IQ6, an atypical isoleucine–glutamine motif in CNGC6. Knockout of *CaM2*, *CaM3*, *CaM5*, and *CaM7* genes led to a marked increase in PM inward Ca^2+^ current under HS conditions; however, knockout of *CaM1*, *CaM4*, and *CaM6* genes had no significant effect on PM Ca^2+^ current [86].In some instances, these motifs function as transcription factors, recognizing and binding to target genes, thereby modulating stress responses in plants.

Several studies have highlighted the critical role of CaMs in plant thermotolerance. In *Arabidopsis*, which possesses nine *CAM* genes, the mRNA levels of these genes were examined in root and shoot tissues from seedlings exposed to normal and HS temperatures. The mRNA levels of all the *CAM* genes, except for *CAM5* in the root and shoot and *CAM1* in the shoot, were upregulated in response to HS treatment [81]. Another study in *Arabidopsis* revealed reduced thermotolerance in knockout mutants of *AtCaM3* after heat treatment at 45 °C for 50 min. Conversely, overexpression of *AtCaM3* in either an *AtCaM3* knockout or wild-type background significantly enhanced thermotolerance. These findings underscore the significance of endogenous AtCaM3 in the Ca^2+^/CaM HS signal transduction pathway [87]. NO also plays a role in thermotolerance in *Arabidopsis* seedlings by acting upstream of AtCaM3. Following HS treatment, NO stimulates the DNA-binding activity of HS transcription factors and the accumulation of HSP18.2 through AtCaM3. This suggests that NO functions in HS signaling and acts upstream of AtCaM3 in thermotolerance, dependent on increased HS transcription factor DNA-binding activity and HSP accumulation [88]. Additionally, overexpression of *CsCaM3*, a *CaM* gene isolated from the cucumber inbred line “02-8,” has been shown to enhance thermotolerance in cucumber plants [89]. That study found that *CsCaM3* transcription was induced by HS or ABA. Overexpression of *CsCaM3* in cucumber plants improved their thermotolerance and protected against both oxidative damage and damage to the photosynthesis system by regulating the expression of HT-responsive genes, including those related to chlorophyll catabolism, under HS stress. In rice, HS was found to cause rapid increases in the cytosolic Ca^2+^ concentration and in the expression and nuclear localization of OsCaM1-1 [90]. These changes are needed to mediate downstream HS-related gene expression, which contributes to the acquisition of thermotolerance in rice. Furthermore, OsCaM1-1 induced the expression of Ca^2+^/HS-related genes such as *AtCBK3*, *AtPP7*, *AtHS factor* (*AtHSF*), and *AtHSP* (even at non-inducing temperatures), and it enhanced thermotolerance in transgenic *Arabidopsis* plants. These findings highlight the significant role of OsCaM1-1 as a mediator of downstream HS signaling.

### 3.2. CMLs in HS Signaling

CMLs differ from CaMs in their length and the number of EF-hand motifs they possess. While CaMs typically contain four EF-hand motifs, CMLs can have varying numbers, ranging from one to six. CMLs are termed “CaM-like” due to their 15% amino acid identity with CaM. These proteins play a crucial role in physiological responses to various stresses, including salinity, drought, heat, and cold. In *Arabidopsis*, 50 CMLs have been identified, compared with 32 in rice [91]. 

In *Arabidopsis*, a rise in external Ca^2+^ or HS significantly increased the mRNA levels of *AtCML12* and *AtCML24* [92]. Both stimuli are known to increase cytoplasmic Ca^2+^, suggesting a role for Ca^2+^ itself in the regulation of CaM-related genes. In rice, another study demonstrated that the expression of *OsMSR2*, a *CML* gene, was strongly upregulated by a wide range of stresses, including cold, drought, and heat, in different tissues and at different developmental stages [93]. The pronounced induction of *OsMSR2* expression by HS suggests its involvement in thermotolerance. Furthermore, *SlCML39*, a *CML* gene from tomato, was found to have a negative impact on thermotolerance during germination and seedling growth in *A. thaliana* [94]. *SlCML39* is expressed in various tissues in tomato plants, including leaves, stems, roots, flowers, and fruits. Overexpression of *SlCML39* in *A. thaliana* resulted in reduced germination rates and compromised seedling growth under HT conditions. Thus, SlCML39 may play a regulatory role in plant responses to HT stress. 

Recent research has significantly advanced our understanding of the roles of CaMs and CMLs as Ca^2+^ sensors in plants. CaMs and CMLs serve as central hubs for integrating various signal transduction mechanisms, allowing cells to respond appropriately to multiple environmental stimuli. Future research efforts are expected to address many of the remaining questions concerning the roles of CaMs and CMLs in plant thermotolerance.

### 3.3. CDPKs in HS Signaling

CDPKs, or CPKs, belong to a class of protein kinases that possess four C-terminal EF-hand motifs, which enable them to sense changes in intracellular Ca^2+^ levels. CDPKs are effector proteins that play pivotal roles in regulating a wide range of physiological processes, including environmental stress responses, in various plant cell types [95]. CDPKs consist of several domains, including a variable N-terminal domain, a serine/threonine protein kinase domain, an autoinhibitory junction domain (JD), and a C-terminal CML regulatory domain (CaMLD) connected by a tether. The CaMLD is composed of four EF-hand Ca^2+^-binding motifs. At low Ca^2+^ concentrations, the C-lobe of a CDPK already contains Ca^2+^ and interacts with the JD, stabilizing its conformation. The JD forms a helical structure that blocks substrate access by being buried within the active site of the kinase domain due to an intramolecular interaction. When the concentration of Ca^2+^ rises, both the N-lobe and C-lobe of the CaMLD interact with the JD, leading to a substantial conformational change that releases the active site [96]. 

The *Arabidopsis* genome contains 34 *CDPK* genes [97], the tomato genome contains 29 *CDPK* genes [98], and the maize genome contains 40 *CDPK* genes [99]. The encoded proteins play important roles in plant growth and development, as well as in environmental stress responses (e.g., HS). In *Arabidopsis*, inactivation of the autoinhibitory domain of AtCPK1 led to increased tolerance to salt, cold, and heat in AtCPK1-transformed *Rubia cordifolia* L. cell cultures, implicating AtCPK1 in the HSR [100]. In tomatoes, LeCPK28 was found to phosphorylate ascorbate peroxidase (APX), enhancing plant thermotolerance [101]. Mutants lacking LeCPK28 exhibited decreased thermotolerance, increased HS-induced accumulation of reactive ROS, elevated protein oxidation levels, and reduced activity of antioxidant enzymes such as APX. Additionally, the tomato *CPK* gene *LeCPK2*, which is predominantly expressed in flowers, exhibited high-level expression at 42 °C [102]. In maize, ZmCDPK7 plays a role in thermotolerance by interacting with and phosphorylating sHSP17.4 at Ser-44, thereby upregulating its expression. ZmCDPK7 can translocate from the PM to the cytosol under HS conditions, and it interacts with the respiratory burst oxidase homolog RBOHB, phosphorylating it at Ser-99 [103]. Another maize *CDPK* gene, *ZmCK3*, exhibited increased transcription in response to drought, salt, and HS in maize seedlings [104]. Overexpressing *ZmCK3* in *Arabidopsis* improved plant survival under conditions of drought and HS. In foxtail millet, *SiCDPK7* was shown to be responsive to extreme temperature stress [105]. *SiCDPK7* overexpression enhanced thermotolerance in both *Arabidopsis* and foxtail millet, with increased transcription of heat and cold stress-responsive genes observed under stressful conditions. In grapevine, the *CDPK* gene *VaCPK29* was shown to be involved in responses to both heat and osmotic stress [106]. *VaCPK29* overexpression in *Vitis amurensis* callus cell cultures and *A. thaliana* plants increased their tolerance to heat and high mannitol stress, indicating that VaCPK29 positively regulates the responses of grapevine plants to these stressors.

The CDPK family is diverse, with multiple members and isoforms across plant species. Recent whole-genome expression analyses have shed light on the transcriptional regulation of CDPKs in response to various stresses, including HS, in important crop species [98,107,108,109,110,111]. These findings provide valuable insight into the functional roles of CDPKs and their potential utility in enhancing the thermotolerance of crops.

### 3.4. CBLs and CIPKs in HS Signaling

CBLs comprise a distinct family of Ca^2+^ sensors in plants with essential roles in Ca^2+^ signaling pathways. Specifically, they interact with and modulate the activity of CIPKs. CBLs bear resemblance to the calcineurin B-subunit in yeast and neuronal Ca^2+^ sensors in animals [112]. CBLs possess four EF-hands, which are Ca^2+^-binding motifs found in various proteins. EF-hands are structural domains characterized by a D-X-D structure, with the loop region serving as the Ca^2+^-binding site. These domains allow proteins to function as Ca^2+^ sensors that can respond to changing levels of intracellular Ca^2+^. In the case of CBLs, the four EF-hand domains facilitate the capture of Ca^2+^ ions, enabling them to play pivotal roles in numerous plant physiological processes [113]. Notably, CBLs lack intrinsic kinase activity. To transmit signals, they must form complexes with CIPKs, which are serine/threonine kinases found in plants. CIPKs possess a functional kinase domain but remain in an inactive state because of the autoinhibition caused by an interaction between the kinase domain and regulatory domain. An inhibitory motif known as NAF/FISL blocks the active site in CIPKs, preventing substrate binding and subsequent phosphorylation. This autoinhibitory mechanism keeps CIPKs in an inactive state until they are activated by binding to a CBL. Once active, the CIPK can regulate downstream proteins. This activation process is crucial for the proper functioning of the CBL–CIPK signaling pathway in plants [114]. When plants encounter stress, such as HS, the intracellular concentration of Ca^2+^ rises. This increase enables Ca^2+^ ions to bind to the EF-hand motifs in CBLs. This binding promotes the interaction of CBLs with the NAF/FISL element in CIPKs. Consequently, the CIPKs become active and can participate in stress response pathways by phosphorylating downstream proteins.

While significant progress has been made in understanding the physiological and biochemical functions of CBLs and CIPKs in plant signal transduction and abiotic stress responses, their roles in thermotolerance are less understood. However, some research indicates their involvement in heat resistance. For instance, in rice, the OsCBL8–OsCIPK17 module plays a critical role in conferring resistance to HS. OsCBL8 facilitates the targeting of OsNAC77 and OsJAMYB by OsCIPK17, leading to enhanced resistance to HT and pathogens in rice [115]. AcCIPK5, a CIPK from pineapple, has been shown to confer salt, osmotic stress, and cold tolerance while negatively regulating the HSR in transgenic *Arabidopsis* plants [116]. 

The CBL–CIPK network is considered a vital regulatory mechanism that deciphers Ca^2+^ signals triggered by HS. Further research will shed more light on the roles of CBLs and CIPKs in regulating plant responses to HS. 

We have included a list of Ca^2+^-binding proteins that are likely involved in plant heat responses (Table 2).

## 4. Ca^2+^ Signaling Networks Mediate the Plant HSR

Ca^2+^ are ubiquitous second messengers in eukaryotes, participating in a wide array of signaling pathways and responses to various environmental conditions. As mentioned above, when plants experience HS, the cytosolic Ca^2+^ concentration rises because of an influx of Ca^2+^ facilitated by membrane-localized Ca^2+^ permeable cation channels. These ions subsequently bind to Ca^2+^-binding proteins, initiating signal transmission to their respective downstream pathways. In recent years, substantial progress has been made in understanding the mechanisms related to thermotolerance in plants. 

The HSR signaling pathways in plants include the Ca^2+^ dependent, ROS, NO, HSF-HSP, HSF-independent, hydrogen sulfide (H_2_S), and unfolded protein response (UPR) pathways, etc. [4,117]. There are interactions and crossovers between different HS pathways. For example, the Ca^2+^ signaling may intersect with the ROS, NO, and HSF-HSP pathways, forming a complex signaling network [4]. Additionally, in the HSF-independent pathway, Ca^2+^ might contribute to the activation of certain transcription factors independent of HSF [118,119]. The H_2_S pathway introduces a novel dimension, suggesting that Ca^2+^ may modulate signaling events in conjunction with H_2_S [120,121]. The association of Ca^2+^ with endoplasmic reticulum (ER) stress responses suggests a potential link between Ca^2+^ and UPR [122,123]. These propositions underscore the versatility of Ca^2+^ signaling and its potential contributions to diverse HSR pathways in plants. Future research is needed to experimentally validate these hypotheses and enhance our understanding of the intricate molecular mechanisms governing plant responses to HS.

In this section, we consider several potential downstream pathways of Ca^2+^ in response to HS in plants (Figure 2).

### 4.1. ROS-Mediated Signaling

In response to HS, plants rapidly accumulate ROS, which are crucial signaling molecules in various stress responses [124]. One particularly potent ROS, H_2_O_2_, is generated in response to the presence of Ca^2+^. H_2_O_2_ is produced during processes such as photorespiration, mitochondrial electron transport, and the beta-oxidation of fatty acids in plants. To maintain cellular homeostasis, the intracellular level of H_2_O_2_ is meticulously regulated through the action of enzymes such as NADPH oxidase, also known as RBOH [125]. Among these enzymes, RBOHD, a ROS-generating NADPH oxidase located at the PM, plays a pivotal role in H_2_O_2_ production. Its activity is directly linked to an increase in cytosolic Ca^2+^ and/or CDPK phosphorylation [126]. Notably, RBOH phosphorylation initiates a positive feedback loop that further enhances both Ca^2+^ and ROS signaling [127]. NADPH oxidase activation results in augmented ROS production in the apoplastic space. These ROS are subsequently transported into the cell through aquaporins, which regulate cellular responses to HS [128]. 

ROS can trigger downstream signaling pathways associated with the HSR in cells through the involvement of multiprotein bridging factor 1 (MBF1), specific HSFs, and MAPKs. This cascade of events can alter a cell’s redox state, primarily through ROS accumulation. MBF1 family proteins operate as transcription co-factors, bridging the gap between transcription factors and the essential transcription machinery. In the context of plants, MBF1 proteins are integral players in abiotic stress responses, particularly in the context of HS [129]. One study found that the accumulation of MBF1c, a member of the MBF1 family, exhibited delayed kinetics in the ROS-producing *rbohd* mutant but accelerated kinetics in the ROS-scavenging *apx1* mutant. This observation underscores the intricate interplay between ROS production and ROS scavenging and highlights the role of MBF1 proteins in modulating the ROS wave in response to HS [130]. Furthermore, MBF1c regulates the expression of more than 30 HS-related transcripts, including *HSFB2a*, *HSFB2b*, and *DREB2A*, which function upstream of HSFA3 [131]. Under HS conditions, members of the HS transcription factor family, including HSFA2, HSFA4a, and HSFA8, translocate rapidly from the cytosol to the nucleus. Importantly, this translocation is redox state-dependent and is mediated by the formation of an intramolecular disulfide bond within these transcription factors. Notably, the formation of this disulfide bond is reversible and is tightly regulated by the cellular redox state [132,133]. These findings reveal a novel mechanism through which plants can swiftly respond to temperature fluctuations by modulating the subcellular localization of key transcription factors.

ROS can also activate the MAPK phosphorylation cascade. In a study conducted in *Arabidopsis* leaf cells, H_2_O_2_ was shown to be a potent activator of MAPKs [124]. Specifically, H_2_O_2_ was able to activate a specific *Arabidopsis* MAPK kinase kinase, ANP1. This activation set in motion a phosphorylation cascade involving two stress-responsive MAPKs, namely, AtMPK3 and AtMPK6 [134]. Researchers were able to enhance the ability of tobacco plants to tolerate HS by overexpressing a constitutively active version of the ANP1 homolog NPK1 [134]. This manipulation suggests that the activation of ANP1 and subsequent activation of MAPK3 and MAPK6 can bolster plant thermotolerance. Furthermore, activated MAPK3 and MAPK6 play crucial roles in enhancing HSP expression. This occurs via the phosphorylation of HSFA2 and HSFA4a [135,136], demonstrating the central regulatory role of MAPK signaling in both the HSR and the expression of genes crucial for cellular protection against heat-induced damage.

### 4.2. NO Signaling

In addition to ROS, another significant free radical produced in plant cells in response to HS is NO. H_2_O_2_ functions upstream of NO in the HS pathway in *Arabidopsis* seedlings. Following HS exposure, the NO levels in seedlings lacking such specific ROS-producing enzymes as *atrbohB*, *atrbohD*, and *atrbohB/D* were found to be lower than those in wild-type seedlings. To mitigate their heat sensitivity, these deficient seedlings were treated with compounds such as sodium nitroprusside or *S*-nitroso-*N*-acetylpenicillamine, which partially rescued their thermotolerance [137]. This observation highlights the interplay between H_2_O_2_ and NO in regulating the plant HSR. Moreover, H_2_O_2_-induced NO was shown to stimulate the activity of antioxidant enzymes, allowing plants to counterbalance excessive H_2_O_2_ levels. This inhibited the DNA-binding activity of HSFs and the accumulation of HSPs. This points to a feedback loop between NO and H_2_O_2_ that regulates thermotolerance [138]. The treatment of seeds with Ca^2+^ enhanced the NO level in *Arabidopsis* seedlings under HS conditions, whereas treatment with EGTA (a Ca^2+^ chelator) reduced it, implicating that CNGC6 stimulates the accumulation of NO depending on an increase in cytosolic Ca^2+^. Western blotting indicated that CNGC6 stimulated the accumulation of HSPs via NO [27]. Additionally, a study found that *Arabidopsis* CaM3 inhibited NO accumulation and enhanced thermotolerance by directly promoting *S*-nitrosoglutathione reductase (GSNOR) activity. This discovery suggests that feedback inhibition occurs between CaM3 and NO in the context of thermotolerance [139]. These findings highlight the intricate regulatory network that controls the response of plants to HS with the help of ROS, NO, and Ca^2+^.

The primary physiological effect of NO is protein *S*-nitrosylation, a redox-based post-translational modification. This modification involves the covalent attachment of a NO molecule to a cysteine thiol on target proteins [140]. Numerous proteins that are integral to the HSR in plants undergo *S*-nitrosylation. This includes various categories of proteins, including HS signaling proteins (CDPK2, CDPK4, CDPK26, CaM, and UVR8), HSPs (HSP70, HSP90, HSP91, HSP88, and HSP60), and enzymes responsible for regulating cellular redox levels (CAT, APX, monodehydroascorbate reductase, SOD, glutathione peroxidase, glutaredoxin, and glutathione *S*-transferase) [4,140]. *S*-nitrosylation regulates various aspects of the modified proteins, including their expression levels, stability, subcellular localization, and enzymatic activity. In doing so, *S*-nitrosylation plays a pivotal role in fine-tuning and coordinating the plant HSR, ensuring the proper functioning of critical proteins involved in stress adaptation and survival.

### 4.3. HSF–HSP Signaling

The synthesis of HSPs represents a crucial protective strategy that enables plants to cope with HS effectively. Within the promoter regions of *HSP* genes, specific sequences known as HS elements (HSEs; 5′-AGAAnnTTCT-3′) are present. These HSEs are recognized and bound by HSFs, which subsequently regulate the expression of *HSP* genes. Ca^2+^ has been implicated in increasing the DNA-binding activity of HSF through direct interactions [141]. 

Studies have provided valuable insight into the role of Ca^2+^–CaM signaling in regulating thermotolerance in plants. Knockout mutants of AtCaM3 or its downstream targets, including AtCBK3 (a protein kinase) and AtPP7 (a protein phosphatase), exhibited decreased HSF activity and reduced HSP synthesis, which impaired the ability of the plants to tolerate HT [87,142,143]. Notably, AtCBK3 and AtPP7 have been reported to interact with HSFs, further underscoring their involvement in the regulation of the HSR. Additionally, overexpression of *OsCaM1-1*, a rice Ca^2+^ sensor, in *Arabidopsis* has been demonstrated to enhance thermotolerance. This enhancement was associated with the elevated expression of HS-responsive genes, including *AtCBK3*, *AtPP7*, *AtHSF*, and *AtHSP* [90]. Furthermore, as mentioned earlier, HSFA2, HSFA4a, and HSFA8 activity are modulated by ROS induced by fluctuations in intracellular Ca^2+^ under HS conditions [132,133]. Notably, AtCaM3 has also been shown to increase thermotolerance via NO-mediated HSF activation and HSP accumulation [88]. Together, these findings highlight the intricate involvement of Ca^2+^–CaM signaling in the regulation of HSF activity and the accumulation of HSPs, and they stress the significance of this pathway in enhancing the ability of plants to withstand HS. 

Plants employ a sophisticated response to HS, activating antioxidant defense systems to neutralize excess reactive ROS and prevent oxidative damage to cellular components. This delicate balance between ROS production and detoxification significantly contributes to enhanced thermotolerance. NO further contributes to this coordinated response by interacting with ROS, influencing their production and scavenging. The crosstalk between NO and ROS enhances the plant’s capacity to resist HS. 

The generation of ROS and NO is triggered in response to the elevation of intracellular Ca^2+^ concentration [126,138]. In a manner similar to CaM, when the intracellular Ca^2+^ concentration rises, both ROS and NO play a contributory role in promoting the synthesis of HSPs through HSF [131]. HSPs act as indispensable molecular chaperones, facilitating proper protein folding and preventing protein aggregation under HS conditions, thereby maintaining cellular homeostasis. Additionally, HSPs contribute to the stabilization of membrane structures, preventing lipid peroxidation and maintaining membrane integrity under HT conditions. This multifaceted response underscores the intricate mechanisms that plants employ to enhance their resilience to HS and safeguard essential cellular functions [144,145]. We have included a list of activated HSPs under the ROS, NO, and Ca^2+^ sensor pathways (Table 3).

## 5. Conclusions and Perspectives

Rising temperatures represent an unusual environmental signal that can substantially reduce global crop yields and pose a significant threat to meeting the future demands of the world’s growing population [2,174]. Elevated temperatures can have a range of physiological, biochemical, and developmental consequences for crops [175]. In response to these challenges, plants have developed intricate signaling networks to detect temperature changes, with the rapid induction of Ca^2+^ signals being one of the earliest responses. This is initiated by an influx of Ca^2+^ into the cytoplasm through PM-situated Ca^2+^ channels. Subsequently, Ca^2+^ is sensed by Ca^2+^-binding proteins, which activate a signal transduction cascade.

Despite significant advancements in understanding the plant HSR over the past two decades, numerous knowledge gaps remain. Among them, the mechanism by which heat is sensed is the “holy grail” of plant thermotolerance research. In past research, several thermosensors have been proposed. For example, it is hypothesized that Ca^2+^ channels located in the PM play a crucial role in perceiving elevated temperatures. HT can enhance the fluidity of the cell membrane, consequently altering the activity of Ca^2+^ channels. Alternatively, a class of receptor kinases specializing in temperature perception may exist. The *Arabidopsis thaliana* genome encodes more than 600 RLKs, with a significant portion being membrane-localized. Many mutants of these RLKs have been found to be heat-sensitive or insensitive [4]. However, the precise roles of the above-mentioned Ca^2+^ channels or RLKs in plant thermosensing and thermoresponding processes require further investigation. 

In addition, the complex interplay of signaling pathways, including Ca^2+^, NO, ROS, and others, deserves further investigation. For instance, Ca^2+^ can activate GSNOR, which produces NO [139], and both Ca^2+^ and NO may influence the generation and scavenging of ROS [132,133]. However, our understanding of this phenomenon is still limited, requiring further investigation.

Furthermore, plants balance their competing requirements for growth and stress tolerance via a sophisticated regulatory circuitry that controls responses to external environments. Therefore, it is desirable to perform in-depth research regarding this balance, such as HSPs synthesis, its impact on energy consumption, and potential trade-offs with processes including photosynthesis, transpiration, and reproductive development. 

Based on the above-mentioned research gaps, future research should prioritize the following areas: (1) Integration of existing pathways. There is a need to integrate various fragmented pathways into a unified primary Ca^2+^ signaling pathway associated with HS. This endeavor could result in a more comprehensive understanding of the role of Ca^2+^ signaling in heat resistance. (2) Identification of new components. Ongoing efforts should focus on identifying new components that sense Ca^2+^ signals induced by HS. This continuing exploration will contribute to an improved understanding of the HSR pathway in plants. (3) Ca^2+^ crosstalk. Understanding the crosstalk between Ca^2+^-mediated HSR and other stress-signaling pathways is essential. This knowledge can shed light on the broader regulatory network that governs plant responses to multiple stressors.

A deeper understanding of the molecular mechanisms may enable the identification of key genes and pathways so as to provide targets for genetic engineering. Moreover, marker-assisted selection, next-generation molecular breeding, precision breeding, and genome editing techniques represent powerful tools to enhance the efficiency of plant breeding programs. These methods will allow breeders to select plants with desirable traits more accurately and quickly and speed up the development of crops better suited to elevated temperatures. These improvements will be conducive to promoting sustainable agriculture.

## Figures and Tables

**Figure 1 ijms-25-00324-f001:**
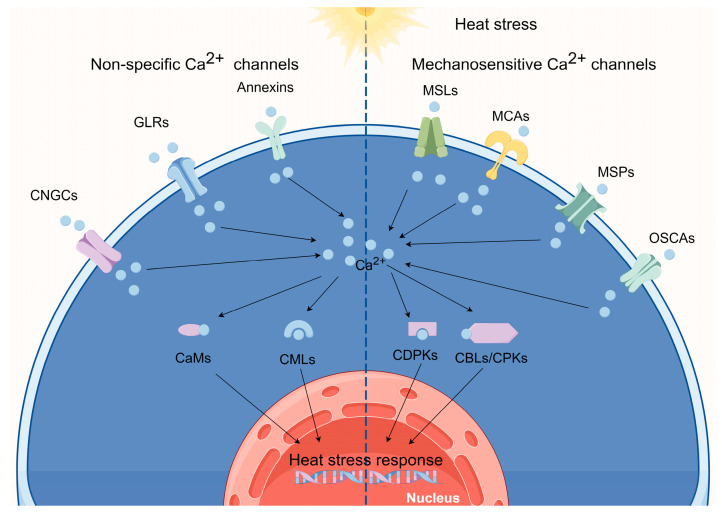
Calcium (Ca^2+^) channels involved in sensing heat. (By Figdraw).

**Figure 2 ijms-25-00324-f002:**
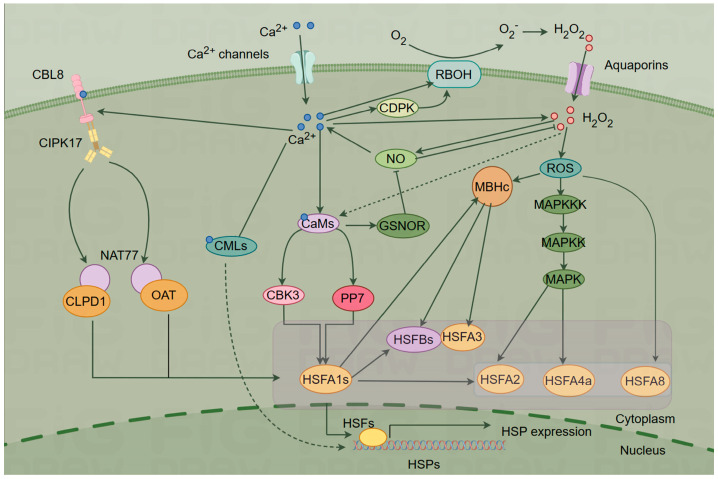
Ca^2+^ signaling pathway under heat shock (HS) (By Figdraw).

**Table 1 ijms-25-00324-t001:** Ca^2+^ channels involved in sensing heat.

Gene Type	Species	Gene Names	Functions	References
Cyclic nucleotide -gated channels (CNGCs)	*Physcomitrella patens*	*CNGCb*	Sensitive to heat stress	[21]
	*Arabidopsis* *thaliana*	*AtCNGC2*	Sensitive to heat stress at the seedling stage; Tolerance to heat stress at the reproductive stage	[21,22,23]
		*AtCNGC4*	Tolerance to extreme temperatures; Response to pathogen infection	[24]
		*AtCNGC6*	Regulates tolerance to extreme temperatures together with hydrogen peroxide (H_2_O_2_) and Nitric oxide (NO)	[25,26,27]
	*Oryza sativa*	*OsCNGC14*	Tolerance to extreme temperatures	[28]
		*OsCNGC16*	Tolerance to extreme temperatures	[28]
Glutamate receptor-like channels (GLRs)	*Arabidopsis* *thaliana*	*AtGLR3.3*	Response to pathogen infection	[43]
	*Vicia faba*	*VfGLR3.5*	Tolerance to drought	[44]
	*Solanum* *lycopersicum*	*SlGLR3.3*	Tolerance to cold stress by regulating apoplastic H_2_O_2_ production and redox homeostasis	[45]
		*SlGLR3.5*	Tolerance to cold stress by regulating apoplastic H_2_O_2_ production and redox homeostasis	[45]
	*Zea mays*	*ZmGLR*	Tolerance to heat stress	[46]
Annexins	*Nelumbo* *nucifera*	*NnANN1*	Tolerance to heat stress	[58]
	*Glycine max*	*GmANN*	Tolerance to high temperatures and humidity stress	[59]
	*Oryza sativa*	*OsANA1*	Tolerance to heat shock, H_2_O_2_ treatment, and abiotic stress	[60]
	*Raphanus* *sativus*	*RsANN*	Tolerance to heat, drought, salinity, oxidation, and ABA stress	[61]
Reduced hyperosmolarity-induced [Ca^2+^] increase channels (OSCAs)	*Zea mays*	*ZmOSCA1.4*	Gene expression increases in response to heat stress	[66]
		*ZmOSCA2.1*	Gene expression increases in response to heat stress	[66]
		*ZmOSCA2.2*	Gene expression increases in response to heat stress	[66]
		*ZmOSCA2.5*	Gene expression increases in response to heat stress	[66]
		*ZmOSCA3.1*	Gene expression increases in response to heat stress	[66]
		*ZmOSCA4.1*	Gene expression increases in response to heat stress	[66]
		*ZmOSCA1.3*	Gene expression decreases in response to heat stress	[66]
		*ZmOSCA1.5*	Gene expression decreases in response to heat stress	[66]
		*ZmOSCA2.4*	Gene expression decreases in response to heat stress	[66]
Mechanosensitive-like channels (MSLs)	*Arabidopsis* *thaliana*	*AtMSL2*	Tolerance to high osmotic stress	[67]
		*AtMSL3*	Tolerance to high osmotic stress	[67]
		*AtMSL8*	Response to PM distortion during pollen grain rehydration and germination	[68]
		*AtMSL9*	Exhibits MS ion channel activity	[69]
		*AtMSL10*	Exhibits MS ion channel activity	[69]
	*Oryza sativa*	*OsMSLs*	Responses to plant growth, development, and various stressors	[70]
Mid1-complementing activity” channels (MCAs)	*Arabidopsis* *thaliana*	*AtMCA1*	Tolerance to cold stress	[73,74]
		*AtMCA2*	Tolerance to cold stress	[73,74]
Piezo channels (MSPs)	*Arabidopsis* *thaliana*	*AtPiezo*	Response to virus infection	[77]

**Table 2 ijms-25-00324-t002:** Ca^2+^-binding proteins that are likely involved in plant heat responses.

Gene Type	Species	Gene Names	Functions	References
Calmodulins (CaMs)	*Arabidopsis* *thaliana*	*AtCaM3*	Tolerance to heat stress	[87,88]
	*Cucumis sativus*	*CsCaM3*	Tolerance to heat stress; Safeguards against oxidative damage	[89]
	*Oryza sativa*	*OsCaM1-1*	Tolerance to heat stress	[90]
CaM-like proteins (CMLs)	*Arabidopsis* *thaliana*	*AtCML12*	Gene expression significantly increased under heat stress	[92]
		*AtCML24*	Gene expression significantly increased under heat stress	[92]
	*Oryza sativa*	*OsMSR2*	Response to cold, drought, and heat stress	[93]
	*Solanum* *lycopersicum*	*SlCML39*	Negative impact on high-temperature tolerance	[94]
Ca^2+^-dependent protein kinases (CDPKs)	*Arabidopsis* *thaliana*	*AtCPK1*	Tolerance to salt, cold, and heat	[100]
	*Lycopersicon* *esculentum*	*LeCPK28*	Tolerance to heat stress	[101]
	*Lycopersicon* *esculentum*	*LeCPK2*	Tolerance to heat stress	[102]
	*Zea mays*	*ZmCDPK7*	Tolerance to heat stress	[103]
		*ZmCK3*	Exhibits increased transcription in response to drought, salt, and heat stress	[104]
	*Setaria italica*	*SiCDPK7*	Response to extreme temperature stress	[105]
	*Vitis amurensis*	*VaCPK29*	Response to heat and osmotic stress	[106]
Calcineurin B-like proteins (CBLs)	*Oryza sativa*	*OsCBL8*	Enhances resistance to high temperatures and pathogens	[115]
CBL-interacting protein kinases (CIPKs)	*Oryza sativa*	*OsCIPK17*	Enhances resistance to high temperatures and pathogens	[115]
	*Ananas comosus*	*AcCIPK5*	Promotes tolerance to salt, osmotic stress, and cold stress while negatively regulating heat stress responses	[116]

**Table 3 ijms-25-00324-t003:** Representative examples of activated HSPs in Ca^2+^ signaling pathway under HS.

	Species	HS proteins (HSPs) Names	References
NO	*Arabidopsis thaliana*	*HSP18.2*	[88]
		*HSP17.7*, *HSP21*	[137,146]
	*Vicia faba*	*Hsp17.6*, *Hsp70*, *Hsp90-1*, and *Hsp101*	[147]
	*Solanum lycopersicum*	*HSP70*	[148]
	*Solanum chmielewskii*		
Reactive oxygen species (ROS)	*Arabidopsis thaliana*	*HSP17.7*, *HSP21*	[26]
	*Solanum lycopersicum*	*HSP40*	[149]
	*Arabidopsis thaliana*	*HSP17.6*, *HSP18.6*	[150]
	*Rosa chinensis*	*HSP17.8*	[151]
	*Zea mays*	*HSP16.9*	[152]
	*Oryza sativa*	*HSP60-B*	[153]
	*Lilium davidii*	*HSP16.45*	[154]
	*Primula malacoides*	*HSP21.4*	[155]
	*Capsicum annuum*	*HSP16.4*	[156]
	*Lilium longiflorum*	*HSP70*, *HSP22.1*, *HSP22.2*, *HSP17.6*, *HSP20*	[157]
	*Glycine max*	*HSP18.5a*	[158]
	*Gossypium hirsutum*	*HSP24.7*	[159]
	*Oryza sativa*	*HSP17.9*	[160]
	*Prunus persica*	*HSP18.5*, *HSP70*, *HSP80*	[161]
	*Oryza sativa*	*HSP80*, *HSP74*, *HSP58*, *sHSPs*	[162]
	*Gossypium hirsutum*	*HSP70-17*	[163]
	*Zea mays*	*HSP17*	[164]
	*Solanum melongena*	*HSP24.1*	[165]
	*Gossypium hirsutum*	*HSP70-26*	[166]
	*Solanum tuberosum*	*HSP70*, *HSP90*, *HSP20*	[167]
	*Festuca arundinacea*	*HSP17.8*	[168]
Ca^2+^ sensor	*Arabidopsis thaliana*	*HSP18.2*, *HSP25.3*, *HSP70*	[25]
	*Arabidopsis thaliana*	*HSP18.2*	[87,88]
	*Gracilariopsis lemaneiformis*	*HSP70s*, *HSP90s*	[168]
	*Arabidopsis thaliana*	*HSP17.6*	[169]
	*Ulva prolifera*	*HSP70*, *HSP90*	[170]
	*Lilium longiflorum*	*HSP101*	[171]
	*Gracilariopsis lemaneiformis*	*HSP70-1*, *HSP70-2*	[172]
	*Dactylis glomerata*	*HSP70*	[173]

## Data Availability

Not applicable.

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
