# Peer review of "Calcium Signaling and the Response to Heat Shock in Crop Plants"

_ijms, 2023, doi:10.3390/ijms25010324_

Round 1

Reviewer 1 Report

Comments and Suggestions for Authors

The paper is a valuable contribution to the field of plant physiology and crop science, especially in the context of climate change. With some enhancements in providing deeper insights, contextualizing within the broader literature, and highlighting future research directions, it could serve as a significant resource for researchers and practitioners in this field.

1. Relevance and Timeliness: The topic of the paper is highly relevant and timely, considering the current challenges posed by climate change and its impact on agriculture. The focus on calcium (Ca2+) signaling in response to heat shock (HS) in plants addresses a critical area of plant physiology and crop science.
2. Comprehensive Coverage: The paper does an excellent job of providing a comprehensive overview of the role of Ca2+ as a second messenger in plants, particularly in the context of heat stress. The detailed discussion on the detection of Ca2+ by various effectors and the subsequent cellular responses is well articulated.
3. Clarity and Organization: The paper is well-structured, making it easy to follow the progression from the general importance of the topic to the specific mechanisms of Ca2+ signaling in response to HS.
Areas for Improvement:
1. Literature Contextualization: While the paper provides a good overview of Ca2+ signaling, it could benefit from a more detailed comparison with other signaling pathways in plants responding to HS. How does Ca2+ signaling integrate with or differ from these other pathways?
2. Experimental Evidence: The paper appears to be a review; however, it would be strengthened by including more specific examples of experimental evidence supporting the roles of Ca2+ in plant thermotolerance. This could include recent studies or meta-analyses that provide quantitative insights.
3. Mechanistic Details: More in-depth discussion on the molecular mechanisms by which Ca2+ signaling alters plant responses to heat stress would be valuable. For instance, how do changes in Ca2+ concentration translate to physiological and genetic changes in plants?
4. Future Research Directions: The paper could benefit from a section discussing potential areas for future research. What are the knowledge gaps in understanding Ca2+ signaling in plants under heat stress? How might future research address these gaps?
5. Global Implications: Expanding on the implications of this research for global agriculture, especially in the context of developing heat-resistant crops, would add value. What are the practical applications of this research in crop improvement and sustainable agriculture?
6. Graphics and Visuals: Consider including more diagrams or visuals that illustrate the Ca2+ signaling pathways and their role in plant responses to HS. Visual aids can greatly enhance understanding, especially for complex biochemical processes.
Concluding Remarks:
The paper is a valuable contribution to the field of plant physiology and crop science, especially in the context of climate change. With some enhancements in providing deeper insights, contextualizing within the broader literature, and highlighting future research directions, it could serve as a significant resource for researchers and practitioners in this field.

Reviewer 2 Report

Comments and Suggestions for Authors

High-temperature is an important environmental factor affecting agricultural production. In this paper, the role of Ca2+ signaling pathways in plant response to high temperature is reviewed, which has good reference value for understanding the mechanism of high temperature tolerance in crops. The review content and level of the paper are suitable for publication in OJMS. Some suggestions are put forward for the writing of the paper. It's recommended to accept after minor revision.

Line 25: “Ca2+ level associated with pollen tube growth” Here, why emphasize the relationship between Ca2+ level and pollen tube growth?

Line 126-158: The full name of “HT” stress or treatment should be written for the first time.

Line 245: It is suggested to change the title of Table 1 to “Ca2+ channels that are likely involved in heat sensing, consistent with the previous statement. The same is in the title of Figure 1.

The second column should be titled " Species ", not " Organism"

Is the title of the third column changed to " Gene names "?

Is the title of the forth column changed to “Functions”

Line 105: “HS tolerance” is mentioned many times, “heat tolerance” is better than HS tolerance”.

Line 417: “HS responses” and “heat responses” were both used. It is recommended to use one of them.

Author Response

High-temperature is an important environmental factor affecting agricultural production. In this paper, the role of Ca2+ signaling pathways in plant response to high temperature is reviewed, which has good reference value for understanding the mechanism of high temperature tolerance in crops. The review content and level of the paper are suitable for publication in OJMS. Some suggestions are put forward for the writing of the paper. It's recommended to accept after minor revision.

Thank you very much. We are delighted to have your approval of our paper.

Line 25: “Ca2+ level associated with pollen tube growth” Here, why emphasize the relationship between Ca2+ level and pollen tube growth?

Thanks a lot for pointing this out. To avoid confusion, we have modified this sentence and indicated the source of the reference.

Line 32 to line 34: “Plants exposed to HT can experience heat shock (HS), which triggers a rise in cytosolic Ca2+ and which disrupts the oscillations in Ca2+ level [3].”

Line 126-158: The full name of “HT” stress or treatment should be written for the first time.

Thank you for pointing this out. We have written the full name when high temperature (HT) is mentioned for the first time.

Line 245: It is suggested to change the title of Table 1 to “Ca2+ channels that are likely involved in heat sensing”, consistent with the previous statement. The same is in the title of Figure 1.

Thanks a lot. We have changed the title of Table 1 to "Ca2+ Channels Involved in Sensing Heat", same to Figure 1.

The second column should be titled " Species ", not " Organism"

Is the title of the third column changed to " Gene names "?

Is the title of the forth column changed to “Functions”

Thank you for pointing these out. We have made the corrections in Table 1 and Table 2.

Line 105: “HS tolerance” is mentioned many times, “heat tolerance” is better than “HS tolerance”.

Thank you so much for this suggestion. We have modified “HS tolerance” to “thermotolerance” throughout the entire manuscript.

Line 417: “HS responses” and “heat responses” were both used. It is recommended to use one of them.

Thank you so much for this suggestion. We have modified “heat responses” to “HS responses (HSR)” throughout the entire manuscript.
